# Lymphocyte Subset Imbalance in Cardiometabolic Diseases: Are T Cells the Missing Link?

**DOI:** 10.3390/ijms26030868

**Published:** 2025-01-21

**Authors:** Francesca Picone, Valentina Giudice, Concetta Iside, Eleonora Venturini, Paola Di Pietro, Carmine Vecchione, Carmine Selleri, Albino Carrizzo

**Affiliations:** 1Department of Medicine, Surgery and Dentistry “Scuola Medica Salernitana”, University of Salerno, 84081 Baronissi, Italy; fpicone@unisa.it (F.P.); ciside@unisa.it (C.I.); pdipietro@unisa.it (P.D.P.); cvecchione@unisa.it (C.V.); cselleri@unisa.it (C.S.); 2Hematology and Transplant Center, University Hospital “San Giovanni di Dio e Ruggi d’Aragona”, 84131 Salerno, Italy; 3Vascular Physiopathology Unit, IRCCS Neuromed, 86077 Pozzilli, Italy; ele.venturini94@gmail.com

**Keywords:** cardiovascular diseases, immune system, immunophenotyping

## Abstract

Cardiometabolic and cardiovascular diseases (CVDs) remain the leading cause of death worldwide, with well-established risk factors such as smoking, obesity, and diabetes contributing to plaque formation and chronic inflammation. However, emerging evidence suggests that the immune system plays a more significant role in the development and progression of CVD than previously thought. Specifically, the finely tuned regulation of lymphocyte subsets governs post-injury inflammation and tissue damage resolution and orchestrates the functions and activation of endothelial cells, cardiomyocytes, and fibroblasts in CVD-associated lesions (e.g., atherosclerotic plaques). A deeper understanding of the immune system’s involvement in CVD development and progression will provide new insights into disease biology and uncover novel therapeutic targets aimed at re-establishing immune homeostasis. In this review, we summarize the current state of knowledge on the distribution and involvement of lymphocyte subsets in CVD, including atherosclerosis, diabetes, hypertension, myocardial infarction, and stroke.

## 1. Introduction

T and B lymphocytes, natural killer (NK) cells, and innate lymphoid cells play pivotal roles in orchestrating immune responses and maintaining immune homeostasis in healthy conditions, as they possess distinct phenotypic and functional activities, enabling appropriate responses to antigens and pathogens [1,2]. In recent decades, our knowledge of lymphocyte functions in immunity has significantly expanded, with innovative studies revealing their involvement in chronic inflammatory conditions, including cardiometabolic and cardiovascular diseases (CVDs) [3]. T lymphocytes represent 10–25% of circulating peripheral blood cells and are mainly divided into two major groups of CD4^+^ and CD8^+^ αβ T cells, along with a small population of γδ T cells (1–5%) and natural killer T (NKT) cells (0.01–0.5%) [2]. CD4^+^ T helper (Th) cells can differentiate into different subsets, each expressing distinct surface molecules, cytokines, and transcription factors, thereby regulating immune responses in diverse ways [4]. For example, Th1 cells promote immune responses against intracellular bacteria and viruses by activating cytotoxic CD8^+^ T cells through the secretion of pro-inflammatory cytokines, such as interferon-gamma (IFN-γ) [5]. Similarly, B cells and innate lymphoid cells (ILCs) have attracted significant attention due to their ability to shape inflammatory responses and influence tissue remodeling processes in physiological and pathological conditions [6].

Identifying and characterizing lymphocyte subsets rely on standard methodologies, including cell sorting, functional assays, and multiparametric flow cytometry (FCM), which is the most commonly used and widely employed technique [7]. Furthermore, FCM can be combined with cell sorting for specific cell subset isolation. Functional assays include measurements of cytokine levels, immune cell proliferation cytotoxic activity, and other effector functions in response to specific stimuli [8]. Molecular biology techniques, such as polymerase chain reaction (PCR) and gene expression analysis, are used to further characterize lymphocyte subsets based on gene expression profiles, up- or downregulation of specific signaling pathways, or specific genetic polymorphisms, which might affect and alter immune responses [9].

CVD remains a leading cause of mortality worldwide, despite increasing awareness to reduce risk factors and the widespread utilization of medications for primary prevention and disease treatment. Emerging evidence identifies the immune system as a crucial player in the development and progression of all types of CVD. It regulates post-injury inflammation and tissue damage resolution through a fine-tuned crosstalk between immune cells, endothelial cells, cardiomyocytes, and fibroblasts. Indeed, exaggerated uncontrolled responses can either lead to excessive tissue damage or fibrosis [10]. This dual role of the immune system underscores the intricate involvement of these cells in the pathogenesis of CVD, which is still not fully understood. For example, CD4^+^ T cells are involved in inflammatory responses in atherosclerosis and hypertension, while CD8^+^ T cells are linked to tissue injury and repair mechanisms during myocardial infarction and stroke [11]. These dynamics highlight the therapeutic potential of targeting T-cell subsets in CVD. NKT cells could also modulate vascular inflammation and influence plaque composition; however, their exact roles, as well as how this regulation may shift in response to different stages of CVD progression, are still under investigation [12]. Recent studies have emphasized the need for comprehensive immune profiling in patients with CVD to better delineate how different lymphocyte populations contribute to disease pathogenesis. The CVD immunome should include specific markers associated with immune cell activation, differentiation, and tissue infiltration, as well as identify molecular pathways that regulate immune cell responses in cardiovascular tissues (Figure 1) [10].

In addition, this intricate relationship between CVD and the immune system is further underscored by the increased risk of cardiovascular complications and mortality observed in autoimmune diseases and Acquired Immune Deficiency Syndrome (AIDS) [13,14]. Despite encouraging preclinical data, a substantial translational effort is required to bridge the gap between bench and bedside and to realize comprehensive T cell characterization in CVD for future T cell-directed therapeutic strategy development.

In this review, we provide a comprehensive overview of lymphocyte contributions in CVD, with a focus on atherosclerosis, hypertension, diabetes, stroke, and myocardial diseases. We also summarize current FCM applications in characterizing and monitoring immune cell subsets in CVD, highlighting their potential as diagnostic biomarkers and therapeutic targets. Flow cytometry immunophenotyping could soon become a powerful diagnostic tool for enhancing the clinical management of CVD patients, improving disease monitoring, and identifying targetable cell populations with novel specific therapies. Indeed, bridging the gap between immunology and cardiology could represent the turning point for advancing our understanding of CVD and for fostering the development of more targeted and effective therapies for these patients.

## 2. FCM Immunophenotyping for T Cell Characterization

Flow cytometry is a powerful technique for immunophenotyping at the single-cell level. It detects scattered light and emitted fluorescence for rapid, precise, and simultaneous measurement of multiple cellular properties, such as size, granularity, and the expression of specific markers [15]. Immunophenotypic characterization of immune cell populations, including T cells, plays a fundamental role in understanding pathogenesis, progression, and response to treatment in immunological conditions, tumors, benign disorders, neurological and infectious diseases, and CVDs [16]. Therefore, accurate standardization of antibody panels and gating strategies is crucial for reliable and reproducible results, especially across laboratories and in routine clinical settings, including trials and research investigations [17].

The introduction of high throughput technologies has enabled the implementation of in-depth immunophenotyping, allowing up to 50 markers that can be simultaneously detected on a single cell, often in combination with molecular features [18]. Multiple standardized panels and gating strategies have been proposed for the identification and characterization of circulating and tissue-specific populations in humans and animals [19]. These panels are usually designed to identify multiple populations within a tube or to deeply characterize a single cell type using the same tube panel, especially for research purposes using technologies that allow the combination of up to 20 different fluorochromes or metals [20]. Despite the promise of full-spectrum flow cytometry, the practical application of >20-color panels in clinical practice remains impracticable, mainly due to the limited availability of diagnostic certified cytometers capable of detecting high numbers of fluorophores. Indeed, clinically-certified cytometers are typically equipped with three lasers, supporting the use of only 10–12 colors. This technological gap between advanced research technologies and clinical practical realities mirrors a diagnostic gap, as complete immunophenotyping of neoplastic cells usually requires multiple panels, also due to the limited number of available fluorophores and their spectral overlaps [21]. Addressing this challenge will require advancements in cytometer technology and the development of standardized, clinically applicable panels to bridge the gap between research and clinical practice.

Standardized flow cytometry panels comprise 8 to 50 markers for comprehensively characterizing the steady-state phenotype of T cell subsets in human peripheral blood and tissues. These panels are designed to identify and characterize T cells, detailing their differentiation and activation status, co-inhibitory receptor expression, and T helper cell subsets, such as CD4^+^ naïve, TEMRA (terminally differentiated effector memory), TEM (effector memory), TSCM (stem cell memory), and TCM (central memory) cells [22,23,24,25,26,27,28,29,30,31,32,33,34,35,36,37,38,39,40,41,42,43,44,45]. For example, naïve T cells (TN) are characterized as CCR7^+^CD45RA^+^CD27^+^CD28^+^CD95^−^, whereas TSCM is CCR7^+^CD45RA^+^CD27^+^CD28^+^CD95^+^. Within the effector memory compartment, early TEM cells are identified as CCR7^+^CD45RA^−^CD27^+^CD28^+^CD95^+^; early-like TEM cells are CCR7^+^CD45RA^−^CD27^−^CD28^+^CD95^+^; intermediate TEM cells are CCR7^+^CD45RA^−^CD27^+^CD28^−^CD95^+^; and terminal TEM cells are CCR7^+^CD45RA^−^CD27^−^CD28^−^CD95^+^. In addition, TEMRA cells are also marked [38,39,40,41,42]. Standardized fluorochromes and clones for T cell identification and characterization, along with their relative optimized panels and gating strategies, are reported in Table 1.

A 50-color panel allows for deep characterization of all T cell subsets and their activation status for research purposes, while this high throughput approach is not possible in clinical settings. Therefore, 10-color panels are the most practical solution, enabling the identification of T cell populations and the evaluation of key activation markers. For CD4^+^ and CD8^+^ immunophenotyping, a panel composed of CCR7(FITC), PD-1(PE), CD45Ra (ECD), CD25 (PC5.5), CD38 (PC7), CD4 (APC), CD8 (APC700), CD3 (APC-750), CD19 (PB), and CD45 (KO) could be used for principal T cell subset characterization (Figure 2).

## 3. Atherosclerosis

Atherosclerosis is a chronic inflammatory condition of the arterial wall that remains a major cause of morbidity and mortality worldwide [46]. Chronic inflammation promotes smooth muscle cell proliferation, extracellular matrix deposition, and the formation of fibrous caps over lipid-rich plaques. However, fibrous caps can break under permanent inflammatory stress, and plaques can rupture or erode (unstable plaques), causing thrombosis and acute cardiovascular events, such as heart attacks or strokes [47]. Macrophages and lipid deposition roles in these events have been extensively investigated; conversely, T cells and NK cells are only currently under the spotlight, with studies showing their involvement in modulating the inflammatory environment within atherosclerotic plaques [48].

NK cells play a dual role in atherosclerosis, acting as both pro-inflammatory and anti-inflammatory agents. They can contribute to plaque instability by killing endothelial and foam cells, and by secreting pro-inflammatory cytokines, like IFN-γ and tumor necrosis factor-alpha (TNF-α), amplifying local inflammation and recruiting other immune cells, such as macrophages [49,50]. NK cells can also regulate immune responses by interacting with dendritic cells (DCs), macrophages, and T cells, to resolve inflammation and stabilize plaques [51].

T cells are now recognized as a major immune cell population in plaques, where they actively contribute to lesion development and progression [52]. T cells are recruited within plaques via endothelial interactions mediated by selectins, integrins, and chemokine signaling, contributing to inflammation through cytokine secretion and clonal expansion [53]. CD4^+^ T cells are a critical component of adaptive immune responses and are involved in various atherosclerotic processes. For example, CD4^+^ TCM maintains immunological memory, coordinates adaptive immune responses, interacts with antigen-presenting cells (e.g., DCs) within atherosclerotic plaques, and contributes to inflammation and immune activation [54]. CD4^+^ TEM participates in local immune response, by producing pro-inflammatory cytokines and interacting with other immune cells (e.g., macrophages) to regulate inflammation and tissue remodeling. In a mouse model, CD4^+^ TEMRA contributes to endothelial-directed cytotoxicity and a pro-inflammatory environment within atherosclerotic lesions, promoting tissue damage and plaque instability [55]. CD4^+^ T cells significantly influence the progression of atherosclerotic plaques, as demonstrated by crossing apoE^−/−^ mice, which are prone to atherosclerosis, with immunodeficient scid/scid mice. The transfer of CD4^+^ T cells from apoE^−/−^ mice into immunodeficient apoE^−/−^/scid/scid mice resulted in larger lesions [56].

On the other hand, in humans, CD8^+^ TCM contributes to chronic inflammatory and plaque progression by producing pro-inflammatory cytokines, like IFN-γ and TNF-α [57,58]. CD8^+^ TEM is involved in local immune response within atherosclerotic lesions, by releasing pro-inflammatory cytokines, participating in cytotoxic responses, and influencing the inflammatory plaque environment. CD8^+^ TEMRA cells have more cytotoxic and pro-inflammatory activities than CD8^+^ TCM, and their presence within atherosclerotic lesions is related to plaque destabilization and rupture [59]. CD8^+^ TEMRA lymphocytes also contribute to local inflammation and exacerbate tissue damage within the arterial wall. Furthermore, elevated frequencies of TSCM cells have been observed in Treg lineage tracker-ApoE^−/−^ mice, suggesting that increased TSCM levels could represent potential hallmarks of advanced atherosclerosis in murine models [60].

The balance and interplay between these T cell subsets in atherosclerosis influence local immune responses, inflammation, plaque stability, and disease progression, and are still under investigation. Indeed, while CD8^+^ TCM cells could maintain surveillance and memory functions in the context of chronic inflammation, CD8^+^ TEMRA lymphocytes are the drivers for plaque destabilization and thrombosis [61]. Therefore, targeting specific CD4^+^ and/or CD8^+^ T cell populations could offer potential therapeutic strategies to modulate immune responses, attenuate inflammation, and mitigate atherosclerosis progression and associated cardiovascular risks [11].

## 4. Hypertension

Hypertension and inflammation are strictly interconnected, as inflammation can contribute to the development and progression of hypertension, and hypertension, in turn, can lead to a chronic inflammatory status. Individuals with hypertension often have elevated levels of pro-inflammatory markers, such as C-reactive protein (CRP) [62], interleukin-6 (IL-6), TNF-α, and reactive oxygen species (ROS) [63,64]. In this condition, subjects experience a long-lasting chronic low-grade inflammation that concurs with endothelial dysfunction, vascular remodeling, and increased vascular tone. Endothelial dysfunction, a hallmark of hypertension, is characterized by impaired vasodilation, increased oxidative stress, and pro-thrombotic status. It promotes vasoconstriction and vascular inflammation, as well as predisposes individuals to hypertensive vascular complications [65,66]. Moreover, inflammation involves several immune cell subsets, such as T lymphocytes, macrophages, and DCs, which infiltrate blood vessels and target tissues, promote inflammation and tissue damage, induce antigen presentation, cytokine release, and immune cell recruitment, and contribute to vascular inflammation and hypertension [67]. In addition, the renin–angiotensin–aldosterone system (RAAS), a key regulator of blood pressure, is intertwined with inflammatory pathways. Angiotensin II, a major RAAS component, exerts pro-inflammatory effects, contributing to vascular inflammation and hypertension. In turn, inflammatory mediators can induce RAAS activity [68].

Anti-inflammatory strategies, such as lifestyle modifications, dietary interventions, pharmacological agents targeting inflammation, and antihypertensive medications with anti-inflammatory properties, have shown promising beneficial effects in mitigating hypertension and associated vascular damage [69].

CD8^+^ T cell subsets can potentially be involved in hypertension-related vascular dysfunction and inflammation; however, the specific roles of each CD8^+^ T cell subset are still unclear. Immune-mediated processes involving CD8^+^ T cells could contribute to endothelial dysfunction, vascular remodeling, and hypertension-related end-organ damage through pro-inflammatory and cytotoxic mechanisms [70]. Longitudinal studies, mechanistic investigations, and clinical trials are needed to elucidate the specific roles of CD8^+^ T cell subsets in the development, progression, and complications of hypertension. In particular, circulating interleukin (IL)-17A-producing CD4^+^ T cells are increased in hypertensive patients, as well as in IFN-γ-producing CD4^+^ and CD8^+^ T cells, independently from angiotensin II levels [71].

Unrevealing the complex interplay between hypertension and inflammation is critical for developing personalized treatment strategies that simultaneously target both conditions. Anti-inflammatory therapeutic approaches could benefit blood pressure control, vascular health, and overall cardiovascular outcomes in hypertensive individuals [72]. Moreover, CD8^+^ T cells could also contribute to these pathogenetic mechanisms. Thus, immunomodulatory agents could improve the clinical management of patients with hypertension [70].

## 5. Diabetes

Recent advancements in the immunological understanding of type 1 diabetes (T1D) are leading to the development of novel therapeutic strategies that could potentially offer effective and safe cures for this disorder [73]. In T1D, the destruction of insulin-producing cells is a consequence of an autoimmune response, with antigen-specific CD8^+^ T lymphocytes playing a pivotal role in this autoimmune response against the pancreas [74]. At disease onset, naïve CD4^+^CD8^+^ T cells are increased, while effector memory subsets and T regulatory cells (Tregs) are decreased. Therefore, the percentage and absolute count of TEM CD4^+^ cells in late-stage disease are the most promising biomarkers of diabetes progression; a therapeutic combination of polyclonal Tregs with IL-2 or rapamycin could improve the survival and functionality of Tregs [75]. Other strategies have investigated combinatorial regimens with systemic therapies, such as anti-CD3, anti-CD20, anti-inflammatory agents, and antigen-specific agents, ultimately leading to antigen-specific Treg induction through the use of tolerogenic peptides and other immune modulators [76].

Type 2 diabetes mellitus (T2DM) is one of the principal risk factors for cardiometabolic disease development and circulating frequencies of immune cells are altered in individuals with diabetes, with a significantly increased risk of complications, including cardiovascular disease and peripheral neuropathy [77]. Moreover, these alterations are associated with insulin sensitivity, glycemic control, and lipid levels. Indeed, T2DM is currently considered a chronic low-grade subclinical inflammatory disease, often associated with metabolic syndrome, excessive nutrient intake, and obesity [78]. Peripheral leukocyte counts are higher compared to T1D and healthy subjects. In particular, T cells isolated from peripheral blood and adipose tissue of T2DM patients exhibit a proinflammatory phenotype, contributing to insulin resistance, pancreatic islet destruction, and disease progression, although their potential role remains controversial [79]. In humans, CD4^+^ T cells also play a pathogenetic role in obesity and insulin resistance; naïve CD4^+^ T cells and Tregs are reduced [80,81] while the number of NK cells is increased. However, NK cells in T2DM are dysfunctional, with low expression of the activation receptors NKG2D and NKp46 and reduced degranulation capacity [82]. Conversely, NKT cells are increased and produce high levels of IFN-γ, IL-4, and IL-17. Additionally, circulating and adipose tissue-resident ILCs, which are augmented, release high levels of IFN- γ [12].

## 6. Stroke

Stroke is the second major cause of death worldwide, after cardiac ischemia, and can result in permanent disability [83]. Innate and adaptive immune responses are actively involved in its pathogenesis; however, their exact roles in stroke recovery remain unclear [84]. During a stroke, neutrophils and monocytes rapidly recruit to the damaged brain tissue, trigger inflammatory responses, and promote debris clearance. Other immune cells, including macrophages, microglia, and T and B cells, are involved in the inflammatory cascade after a stroke [85]. Microglia, the brain-resident immune cells, are activated within minutes after an ischemic event and can release both pro- and anti-inflammatory cytokines, orchestrating immune responses to prevent long-term tissue damage [86,87,88]. Indeed, while inflammation is necessary for tissue repair, excessive responses can exacerbate damage [89], resulting in blood–brain barrier (BBB) disruption, and leading to increased neuronal death and impaired neurogenesis [90]. Moreover, chronic inflammation can extend into subacute and chronic phases of stroke recovery, contributing to post-stroke cognitive decline and further impairing functional outcomes. For instance, elevated levels of pro-inflammatory cytokines, such as TNF-α and IL-6, are associated with worse outcomes [91,92].

CD4^+^ T cells can influence post-stroke inflammation [93]; Th1 cells produce IFN-γ and other cytokines and promote inflammation, potentially aggravating brain injury, while Th2 cells mediate anti-inflammatory effects through IL-4 and IL-13 secretion, favoring tissue repair [94]. Tregs are also critical in controlling excessive inflammation and promoting recovery [95,96] by releasing anti-inflammatory cytokines, such as IL-10 and transforming growth factor (TGF)-β. Antigen-specific CD8^+^ T cells can recognize and kill neurons, contributing to further neuronal loss [97,98]. B lymphocytes might contribute to neuroinflammation through antibody production, although their exact role in brain injury during a stroke has not been well-established [16].

Novel therapeutic approaches include immune response targeting [99], which involves inhibiting neutrophil and M1 macrophage infiltration, inducing anti-inflammatory Tregs and M2 macrophages, and regulating cytokine production, to limit secondary damage and promote neuroprotection. Other therapies aim to maintain BBB integrity and recovery to reduce leukocyte infiltration and mitigate the detrimental effects of inflammation [100].

## 7. Myocardial Diseases

The mammalian heart is composed of striated myocytes, which enable rhythmic contractions throughout the lifespan, and includes both resident and circulating immune cells [101]. It plays a pivotal role in maintaining cardiac homeostasis and facilitating adaptive responses to injury.

Macrophages are the majority of resident immune cells in cardiac tissue, accounting for ~40% of CD45^+^ cells, and express CD64, CD163, MERTK (MER proto-oncogene tyrosine kinase), LYVE1 (lymphatic vessel endothelial hyaluronan receptor 1), and MHC class II [102]. Following ischemic myocardial injury, both resident and monocyte-derived macrophages rapidly engulf endogenous debris, and CD14^+^CD16^−^ monocyte frequencies are associated with adverse cardiac remodeling and function [102].

Lymphocytes also play an important role in myocardial diseases, with T cells being essential for maintaining cardiac homeostasis and mediating the heart’s response to injury [103]. T cells are involved in fetal development and functions of the heart, including protecting against infections and other insults, cardiac remodeling (a process by which the heart adapts to injury), and blood vessel formation [104]. Indirect evidence of the involvement of T cells in maintaining cardiac functions involves the development of myocarditis under immune checkpoint inhibitors (ICIs) for cancer treatment, likely induced by programmed cell death (PD)-1 activation of antigen-specific T cells against PD-1 ligand (PD-L1)-expressing cardiac cells [105]. During ICI-induced myocarditis, CD8^+^ TEMRA cells expand and express CXC motif chemokine receptor 3 (CXCR3), which directs them to the site of inflammation, such as the heart. In addition, Tregs exert anti-inflammatory effects through the secretion of immunosuppressive cytokines, such as TGF-β and IL-10, and influence antigen-presenting cells and CD8^+^ effector T lymphocytes, leading to decreased inflammation and mitigation of immune reactions [106]. During myocardial injury, Tregs limit inflammation, reduce fibrosis, and improve cardiac function by modulating effector T cells and macrophage activities [106]. Because of their role in promoting the resolution of inflammation and supporting tissue repair, Tregs represent a potential therapeutic target in the treatment of myocardial diseases, such as ischemic injury and heart failure.

B cell roles in myocardial diseases are less studied, and most of the available data are derived from murine model studies. In the myocardial infarction model, CD19^+^ B cells are increased in the myocardium and pericardial adipose tissue following ischemia, peaking between 5 and 7 days after injury, and their depletion results in improved cardiac function [107]. B lymphocytes negatively influence left ventricular (LV) remodeling by recruiting Ly6C^+^ monocytes from the bone marrow [108]. Simultaneously, intracardiac injections of bone marrow-derived B cells protect the myocardium from cardiomyocyte apoptosis, highlighting the dual role of this immune cell population. Indeed, B cells can limit inflammation and promote tissue repair by producing regulatory cytokines, such as IL-10 [109]. However, under chronic inflammatory conditions, these cells exacerbate myocardial damage by promoting autoimmunity through heart-specific autoantibody production or by driving maladaptive immune responses [108]. In human hearts, B cells are present at similar levels to CD4^+^ and CD8^+^ T cells; however, in the mice model, during acute myocardial infarction, a temporary decrease in circulating B cells is observed, followed by an increase 24 h after reperfusion [109].

NK cells are another immune cell population involved in myocardial homeostasis and response to injury, especially after ischemic events [110]. NK cells can produce large amounts of pro-inflammatory cytokines, such as IFN-γ, contributing to adverse remodeling after myocardial infarction [111]. However, there is discordant evidence suggesting that NK cells might play a role in limiting excessive inflammation and promoting tissue healing [111].

DCs are professional antigen-presenting cells that populate the myocardium, where they can uptake, process, and present cardiac antigens to T cells after tissue damage, such as under injury or infections. During myocardial infarction or myocarditis, DCs can drive protective immune responses or contribute to pathological inflammation and cardiac dysfunction, based on predominant stimuli [112].

Fibroblasts actively participate in immune response within the myocardium. This is due to their crosstalk with macrophages and T cells, which finely modulate cardiac remodeling processes, resulting in functional recovery or heart failure. Therapeutic targeting of signaling pathways involved in fibroblast-immune cell crosstalk, such as TGF-β or IL-1 downstream, could offer new treatment approaches to prevent maladaptive fibrosis in myocardial diseases [113].

## 8. CVD Pharmacological Treatment and Imbalance of Lymphocyte Subsets

While there is growing evidence of the pivotal role of the immune system in cardiometabolic disease, there are few immune-based therapies for CVD treatment, with a limited number of clinical trials investigating the role of Tregs in atherosclerosis, such as LeukoCAPE-2 trial, NCT03105427, an observational case-only study. In this study, Tregs were studied as predictors of cardiovascular risk in two patient groups: those with a known CVD history undergoing major non-cardiac surgery, and those who have undergone cardiovascular surgery [114]. Results from the LeukoCAPE-2 trial indicate that preoperative Treg levels can highly and specifically increase preoperative risk assessment in combination with other well-established cardiac biomarkers, especially in intermediate-risk patients for major adverse cardiac and cerebrovascular events (MACCE) [115]. Conversely, in the CANTO study, IL-1β neutralization was investigated in CVD patients, showing a significant reduction in non-fatal myocardial infarction, non-fatal stroke, or cardiovascular death, although the mechanism remains unclear [116].

Clinical data supporting the key role of T-cell modulation in cardiovascular disease are still emerging. One randomized trial investigated canakinumab (IL-1β monoclonal antibody inhibiting the activity of Tregs) in patients with a history of myocardial infarction, demonstrating a significant reduction in the recurrence of cardiovascular events [115]. A randomized clinical trial involving patients with stable ischemic heart disease and acute coronary syndrome used low-dose IL-2. Aldesleukin increased the average circulating Treg levels, alleviated endothelial dysfunction, and reduced the formation of atherosclerotic plaques, favoring cardiovascular health [114,117].

Moreover, numerous studies have investigated the roles of different lymphocyte subsets in CVD across various tissues and species, as detailed and summarized in Table 2.

## 9. Conclusions

Comprehensive lymphocyte subset immunophenotyping in clinical settings represents a long-standing area of interest, yet its clinical application remains limited due to the following substantial technical and logistical challenges: (i) the rarity of certain subpopulations, which reduces the sensitivity and specificity of FCM for their quantification and requires optimized antibody panels, rigorous gating strategies, and large sample volumes, which are not always available, especially in pediatric or critically ill patients; (ii) inter-operator and inter-laboratory variabilities, such as sample handling, staining protocols, and instrument calibration, which introduce discrepancies and, thus, reduce results comparability; (iii) pre-analytical variables, such as sample storage conditions and processing times, which add other variations that affect rare subpopulation quantification. Therefore, protocol standardization is essential to establish international consensus criteria for processing, analysis, and reporting, ensuring more reliable and comparable results, especially in clinical settings, including CVD.

Despite these challenges, integrating immune profiling into the diagnostic and therapeutic landscape of CVD holds great promise. A better understanding of the intricate crosstalk between immune cells and disease pathology could enhance prognostic accuracy and lead to the development of targeted therapies, such as selective modulation of Tregs. Furthermore, identifying immune-based biomarkers for early disease detection and therapy response prediction represents a promising frontier. While animal models have significantly contributed to advancing methodological approaches, their species-specific differences warrant careful extrapolation of findings to human disease. Consequently, refining cytometric techniques and identifying biomarkers will be pivotal in overcoming current limitations and unlocking the full potential of lymphocyte subpopulation analysis for improving the clinical management of CVD.

## Figures and Tables

**Figure 1 ijms-26-00868-f001:**
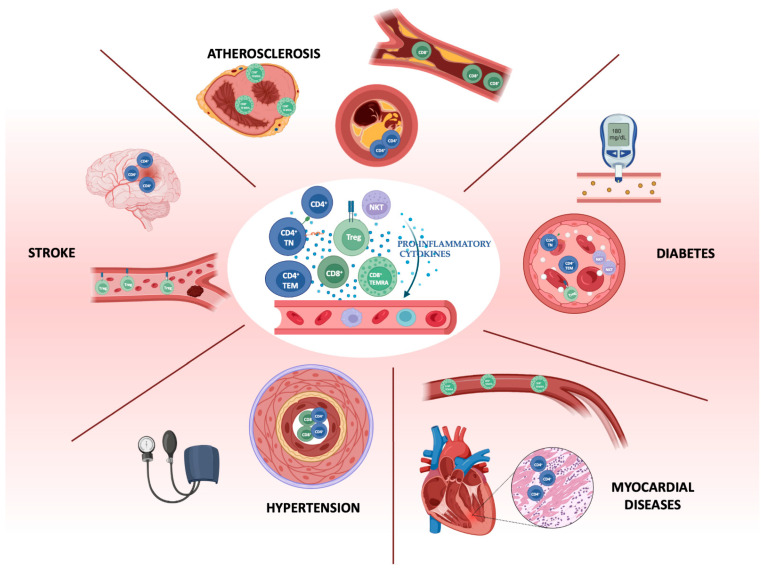
A schematic summary of T cell subset involvement in the pathophysiology of cardiovascular disease in humans. This figure was produced using the free version of BioRender.

**Figure 2 ijms-26-00868-f002:**
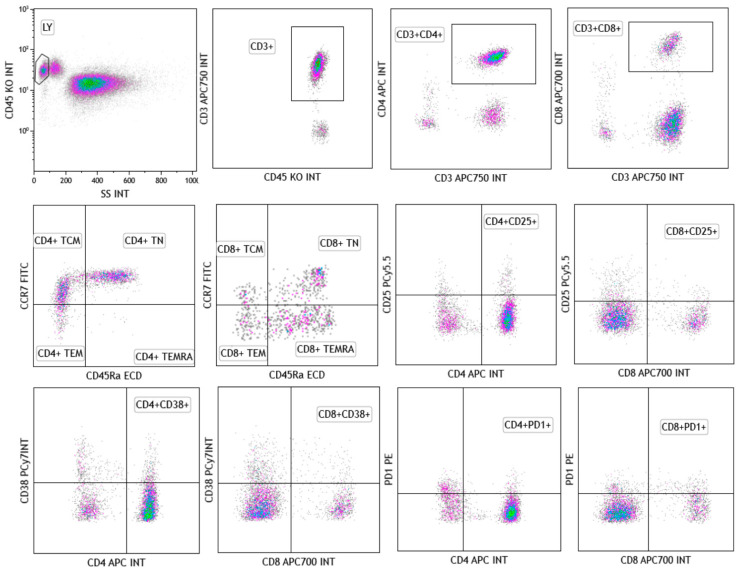
A gating strategy for T cell subset identification. Lymphocyte populations were first identified based on linear parameters (side scatter area, SSC-A) and CD45 expression; cells were then gated for CD3 expression. CD3^+^CD4^+^ and CD3^+^CD8^+^ populations were further identified and analyzed for CD197 (CCR7) and CD45RA expression, for gating of effector memory (CCR7⁻CD45RA⁻), central memory (CCR7^+^CD45RA⁻), T naïve (CCR7^+^CD45RA^+^), and terminally differentiated effector memory re-expressing CD45RA T cells (CCR7⁻CD45RA^+^). Lymphocyte populations were further studied for activation marker expression, such as CD25, CD38, and PD-1.

**Table 1 ijms-26-00868-t001:** T cell markers and published standardized antibodies and gating strategies for human T cell subset immunophenotyping.

Specificity	Fluorochromes	Clones	Purpose	OMIP
CD2	PerCP/Cy5.5	TS1/8	Identification of ILCs, NK cell phenotyping	69 [31], 109 [45]
	APC-AF700	39C15		81 [36]
	Qdot 605	S5.5		102 [43]
CD3	AF594, ECD, QD605, AF488, BV570, BUV496, AF647, APC-AF750, BUV395	UCHT1	Mature T cells	17 [22], 24 [24], 30 [25], 42 [26], 63 [29], 67 [30], 101 [42], 102 [43], 80 [35], 81 [36], 91 [39]
	V500, APC-Cy7	SP34-2		23 [23], 78 [34]
	AF700	HIT3a, UCHT1		90 [38]
	PerCP/Cy5.5, APC-H7, BV510, Spark Blue 550	SK7		53 [27], 60 [28], 71 [32], 69 [31], 99 [41], 109 [45], 94 [40]
	BUV805	SK3		71 [32]
	BV605	OKT3, UCHT1		77 [33], 106 [44]
	BV711	MM1A		89 [37]
CD4	QD800	OKT4	CD4 T cell and NKT-like cell lineage marker	17 [22]
	APC-H7, BUV395, BV510, BUV496, BUV661, BUV805, cFluor YG584, PE-Cy5.5, PerCP, NovaFluor Blue 610-70s, BV480, NovaFluor Blue 585, PerCP/Fire 806	SK3		23 [23], 24 [24], 90 [38], 30 [25], 42 [26], 53 [27], 60 [28], 63 [29], 67 [30], 69 [31], 78 [34], 94 [40], 99 [41], 101 [42], 102 [43], 109 [45]
	BV750	RPA-T8		71 [32]
	BV570, BUV496	RPA-T4		80 [35], 106 [44], 91 [39]
	PE-Cy7	SFCI12T4B11		81 [36]
CD5	PE-Cy5.5	BL1a	T-cell lineage, B-cell subset	81 [36]
CD7	PE-Cy7	M-A251	Naïve/effector T cell	71 [32]
	FITC	8H8.1		81 [36]
CD8	QD585, eF650, BUV496, PE, BUV563, BUV805, BV570	RPA-T8	CD8 T, NK, NKT-Like, and MAIT cells	17 [22], 30 [25], 63 [28], 67 [30], 78 [34], 80 [35], 101 [42]
	PerCP-Cy5.5, BV510, BUV805, BV750	SK1		24 [24], 60 [28], 69 [31], 99 [41], 109 [45], 91 [39]
	BV650	RPA-T8, SK1		42 [26], 53 [27]
	BV786	NCAM16.2		71 [32]
	PB	B9.11		81 [36]
	FITC	SK1, RPA-T8, B9.11		94 [40], 106 [44], 23 [23]
	NovaFluor Blue 585	OKT8		102 [43]
	PE-Texas Red	CC63		89 [37]
CD11a	BUV496	HI111	Activation	99 [41]
CD16	PE-Cy7, APC-Cy7, BUV496, BV650, AF700, BV605, PE-AF700	3G8	T-cell subsets, NK cells, monocyte differentiation, eosinophil exclusion, myeloid lineage	23 [23], 24 [24], 42 [26], 69 [31], 109 [45], 78 [34], 80 [35], 94 [40], 101 [42], 102 [43]
	APC-efluor780	ebioCB16		77 [33]
	PE	B73.1		81 [36]
CD25	ECD	B1.49.9	IL-2 Receptor a, Treg, lymphocyte activation marker	23 [23]
	BV421	M-A251, HIL-7R-M21		24 [24], 71 [32]
	BV711, BUV563, BYG584, BUV737	2A3		30 [25], 42 [26], 60 [28], 63 [29], 80 [35]
	PE/dazzle™ 594, BUV805	M-A251		53 [27], 106 [44]
	BB700, PE-Cy7, BV605, BB515	BC96		67 [30], 94 [40], 99 [41], 102 [43]
	cFluor BYG710	4 E 3		69 [31], 109 [45]
	PE-Cy5	IL-A111		89 [37]
	PE/CF594	M-A251, BC96		90 [38], 91 [39]
CD27	BV785, BV750	O323	Memory B cells; naïve and CM CD4^+^ and CD8^+^ T	67 [30], 94 [40]
	BV786	L128		60 [28]
	BB515, PE-Cy7, BUV615, PB, BUV805, BB660	M-T271		63 [29], 109 [45], 78 [34], 80 [35], 99 [41], 101 [42], 102 [43]
CD28	BV650, BV711, BV750, BV480	CD28.2	T cell and NK cell differentiation, co-stimulation molecule	69 [31], 109 [45], 90 [38], 99 [41], 102 [43]
	PE	CD28.2, DX2		60 [28], 71 [32]
CD31	cFluor YG584, BV786	WM59	Differentiation, adhesion molecule	67 [30], 80 [35]
CD38	PE-Cy5.5	LS198.4.3	Monocyte, mDC, T cell, and B cell activation/differentiation, plasmablasts	23 [23]
	PE-Cy5, BV421, BUV661, APC/Fire810, BB660, BB700, BV480	HIT2		24 [24], 42 [26], 77 [33], 67 [30], 69 [31], 78 [34], 101 [42], 106 [44]
	eF450, PerCP-eF710, APC/Fire810, BUV737, BUV395	HB7		30 [25], 94 [40], 102 [43], 109 [45], 53 [27]
	Spark YG 581	S17015F		99 [41]
CD39	BUV661	TU66	B cell, T regulatory, Treg activation marker and monocyte differentiation	60 [28], 69 [31], 109 [45]
	PE-Cy7, PE/Fire810, BV785, R718, APC/Fire750	A1		80 [35], 94 [40], 99 [41], 102 [43], 53 [27], 106 [44]
CD45	PB, KO	J.33	Pan-leukocyte antigen	23 [23], 81 [36]
	PerCP, Spark Blue 550	2D1		69 [31], 109 [45], 99 [41]
	APC-R700	UCHT1		71 [32]
	BV480, BUV496, BV570, BV785, BUV805, BUV395	HI30		77 [33], 78 [34], 94 [40], 101 [42], 102 [43], 106 [44]
CD45RA	QD655	5H9	Memory/differentiation, TEMRA T cells, naive/memory	17 [22]
	BV650, BV570, BV711, PerCP-Cy5.5, PB, BUV496, APC-eFluor 780, Spark UV 387	HI100		24 [24], 30 [25], 67 [30], 53 [27], 42 [26], 80 [35], 94 [40], 90 [38], 91 [39], 101 [42], 102 [43]
	BUV395	HI100, 5H9		63 [29], 69 [31], 99 [41], 109 [45]
CD45RO	BV570, PerCP, PB	UCHL1	Phenotyping of T cells, naive versus memory, memory CM and EM	60 [28], 78 [34], 102 [43], 99 [41], 106 [44]
	PE-CF594	DX12		71 [32]
	FITC	IL-A116		89 [37]
CD49d	BUV563	L25	Activation	99 [41]
CD56	PE-Cy7	N901 (NKH-1)	N-Cam, NK, NKT, γδ T cell differentiation, and some memory CD8^+^ T cells	23 [23]
	BV605, BV510, BV711	HCD56		24 [24], 94 [40], 101 [42]
	BUV737, BB790, BUV737, PE-Cy7, PE-CF594, PE, BV786	NCAM16.2		42 [26], 63 [29], 69 [31], 77 [33], 80 [35], 81 [36], 99 [41], 106 [44]
	BV650	M-A251		71 [32]
	PE-Cy5	B159		78 [34]
	BUV563	NCAM16.1		102 [43]
	cFluor YG584	5.1H11		109 [45]
CD57	FITC, BB515, PE	NK-1	NK and CD8^+^ T cell immune senescence, Terminal differentiation, T cell and NK cell differentiation, and memory status	24 [24], 78 [34], 101 [42], 102 [43]
	cFluor B532, APC	HNK-1		69 [31], 94 [40]
	BV480	M-T701		71 [32]
	BV605	QA17A04		80 [35]
	cFluor B532 BV785	cFluor B532 BV785		109 [45]
CD62L	BUV395, BUV496, BV650	DREG-56	T-cell, TN and TCM cells	80 [35], 106 [44], 94 [40]
	PerCP-Cy5.5	CC32		89 [37]
CD69	PE	TP1.55.3	Phenotyping of T cells, tissue residency marker, activation	23 [23]
	BV711, BUV496, BUV737, PE/Fire640, BV650	FN50		60 [28], 80 [35], 91 [39], 94 [40], 102 [43]
	BB790	FN50, 2E7		67 [30], 71 [32]
CD73	BUV661	AD2	Subset differentiation	99 [41]
CD94	APC/Fire750	DX22	Activation marker on NK and CD8 T cells	94 [40]
CD95	BUV737, PE-Cy5, BV650, BUV737, PE/Fire640	DX2	T cell activation and differentiation	60 [28], 69 [31], 109 [45], 80 [35], 90 [38], 99 [41]
	BUV661	NK-1		71 [32]
CD103	BB630	UCHL1	Intraepithelial lymphocytes and Treg	71 [32]
	BV750, PE-Fire640	Ber-ACT8		102 [43], 109 [45]
CD112	PE	TX31	Inhibitory ligand	106 [44]
CD122	BV421	TU27	IL-15 receptor subunit on T cells, NK, and NKT-like cells	106 [44]
CD127 (IL-7Rα)	APC-AF700	R 34.34	IL-7 Receptor a, Treg, Tregs/memory/differentiation, and ILC identification	23 [23]
	APC, BV421, cFluor R720, AF647, AF700, PE-Cy7	A019D5		24 [34], 53 [27], 69 [31], 109 [45], 94 [40], 99 [41], 67 [30]
	APC-eF780	RDR5		30 [25], 42 [26]
	BB700, BB630, BV786, APC-R700, RB744	HIL-7R-M21		63 [29], 78 [34], 90 [38], 91 [39], 102 [43]
	PE-Cy5	OF-5A12		71 [32]
	PE-Cy5.5	eBioRD5		101 [42]
CD137	BB790, AF647	4B4-1	Activation marker	60 [28], 91 [39]
CD154 (CD40L)	BV480	TRAP1	Activation marker	60 [28]
	BV421, BV605	24–31		91 [39], 67 [30]
CD155	BUV737	SKII.4	Inhibitory ligand	106 [44]
NKG2A (CD159a)	BB790, BV421	131,411	NK and NKT-like cell inhibitory receptor	106 [44], 80 [35]
	APC	REA110		69 [31]
NKG2C (CD159c)	AF700	134591	NK cell-activating receptor	24 [24]
	PE	FAB138P, REA205		80 [35], 69 [31]
CD161	FITC, BV786, BV650, BV421	DX12	MAIT, NK, NKT, and a subset of CD8^+^ T cells	17 [22], 63 [29], 101 [42], 102 [43]
	PerCP-Cy5.5, BV785, APC	HP-3G10		30 [25], 94 [40], 109 [45]
	BUV737	CD28.2		71 [32]
	PE-Vio770	REA631		91 [39]
CD183 (CXCR3)	FITC, PE, BV421, APC	1C6	Dendritic cell, T cell, and B cell differentiation, migration, Tfh/Th marker	30 [25], 42 [26], 63 [29], 91 [39]
	PE-Fire 640, PE-Cy7, PE-Dazzle 594	G025H7		102 [43], 69 [31], 109 [45], 99 [41], 67 [30]
	PE-Cy5	1C6/CXCR3		17 [22], 80 [35], 90 [38]
	BV510	12G5		71 [32]
CD184 (CXCR4)	BUV563	RF8B2	Th marker	71 [32]
CD185 (CXCR5)	BV750, PE-Cy7, AF647, BUV805, BB700, BUV563	RF8B2	mNKT/MAIT cells, T cell differentiation, Th subset, Tfh cells	69 [31], 109 [45], 42 [26], 17 [22], 91 [39], 99 [41], 67 [30]
	BUV615	RF8B2, 13B 1E5		63 [29], 71 [32]
	PE	J252D4		90 [38]
CD186 (CXCR6)	BB700	1G1	Th marker	71 [32]
CD194 (CCR4)	PE-Cy7	TG6/CCR4	Chemokine receptor; Th subset identification	17 [22]
	BUV395	11A9		71 [32]
	BV605	L291H4, IG1		90 [38], 91 [39], 42 [26]
	BUV615, BB700, BV786, PE	1G1		80 [35], 99 [41], 102 [43], 30 [25]
CD195 (CCR5)	BUV563	2D7/CCR5	Monocyte, dendritic cell, T and B cells	109 [45]
	PE/Cy7	J418F1		90 [38]
CD196 (CCR6)	BV786	G034	Chemokine receptor, Th subset, Th17 cells, differentiation/trafficking	30 [25]
	BV711	BV711, G034E3		69 [31], 109 [45]
	BV605, BV650, BV785, BV711, APC	G034E3		17 [22], 91 [39], 42 [26], 99 [41], 90 [38]
	BB630, BUV563	11A9		63 [29], 80 [35]
	BUV496	2-L1-A		71 [32]
CD197 (CCR7)	Ax680, BUV395, PE-CF594	150503	Naïve/memory classification, TN and TEM cells	17 [22], 60 [28], 101 [42], 30 [25], 42 [26]
	BV605, BV785, BV421, PE, PE-Cy7, PE-Fire810, BV650, AF700	G043H7		102 [43], 24 [24], 90 [38], 53 [27], 63 [29], 69 [31], 109 [45], 91 [39], 99 [41], 67 [30]
	APC-Cy7	1D11		71 [32]
CD200	BB660	MRC OX-104	Inhibitory ligand	106 [44]
CD226 (DNAM-1)	BV421	11A8	Exclusion of functionally unstable Treg for in vitro expansion, tTreg identification	99 [41]
	Real Blue 780, BUV563, BV750	DX11		109 [45], 106 [44], 80 [35]
	PE/Cy7	HB7		53 [27]
CD272	RB613	J168-540	Activation	102 [43]
CD274 (PD-L1)	PE-CF594	MIH1	Inhibitory ligand	106 [44]
CD275	PE-Cy7	2D3	Co-stimulatory ligand	106 [44]
CD276	BUV615	7-517	Inhibitory ligand	106 [44]
CD278	PE-Cy5.5	ISA-3	Phenotyping of T cells, activation	102 [43]
	BB630	DX29		106 [44]
CD279 (PD-1)	PE-Cy7, PE-CF594, BUV661, BV786, BUV615	EH12.1	T cells, activation and T cell inhibitory receptor, exhaustion marker, Tfh marker	102 [43], 63 [29], 80 [35], 106 [44], 78 [34], 91 [39]
	BV785, BV421, BB660	EH12.2H7		69 [31], 17 [22], 94 [40], 67 [30]
	BB515	G46-6		71 [32]
	PE	NAT105		99 [41]
CD366 (TIM-3)	BB515	A15153G, 7D3	T cell inhibitory receptor	99 [41], 109 [45]
	BV480	7D3		80 [35]
	PE-Cy5	F38-2E2		106 [44]
FoxP3	APC	236A/E7	Treg identification	53 [27]
	PE-Cy5.5, AF660	PCH101		60 [28], 80 [35], 106 [44], 99 [41]

Abbreviations. Th, T helper; Treg, T regulatory cells; DC, dendritic cell; TEM, effector memory T cells; TCM, T central memory; TN, T naïve; TEMRA, terminally differentiated effector memory T cells re-expressing CD45RA; NK, natural killer; ILCs, innate lymphoid cells, Tfh, Th follicular cells; MAIT, mucosal-associated invariant T cells.

**Table 2 ijms-26-00868-t002:** Species, diseases, and cell subset involvement in the published literature.

**Human**			
**Disease**	**Cell Subset**	**Tissues**	**References**
Atherosclerosis	CD8^+^ CD8^+^ TEMRA CD4^+^	Endomyocardial biopsy Blood Atherosclerotic plaque	Friebel, J. et al. [57] Grivel, J.-C. et al. [59] Fan, L. et al. [11]
Hypertension	CD4^+^ and CD8^+^	Blood	Itani, HA. et al. [71]
Diabetes	CD4^+^ TEM CD4^+^ TN Treg NKT	Blood	Teniente-Serra, A. et al. [75] Nekoua, M.P. et al. [80] Jagannathan-Bogdan, M. et al. [81] Daryabor, G. et al. [12]
Stroke	CD4^+^ Treg	Brain Blood	Jin, W.N. et al. [93] Wang, H. et al. [98]
Myocardial diseases	CD8^+^ TEMRA CD4^+^	Blood Ventricular tissue	Zhu, H.A.-O. et al. [91]
**Mouse**			
**Disease**	**Cell Subset**	**Tissues**	**References**
Atherosclerosis	CD4^+^ TEMRA CD4^+^ CD8^+^	Heart cryosections Blood	Delgobo, M. et al. [55] Zhou, X. et al. [56] Padgett, L.E. et al. [60]
Hypertension	CD8^+^	Blood	Hengel, F.E. et al. [70]
Stroke	CD4^+^ T cells	Spleen Brain	Jin, W.N. et al. [93]
Myocardial diseases	CD8^+^ TEMRA CD4^+^, CD8^+^ CD4^+^ CD19^+^CD5^+^	Blood Heart Pericardial adipose tissue	Zhu, H.A.-O. et al. [91] Wu, L.A.-O.X. et al. [83]
**Rat**			
**Disease**	**Cell Subset**	**Tissues**	**References**
Myocardial diseases	Breg	Hearts	Huang, F. et al. [109]

Abbreviations. Tregs, T regulatory cells; TEM, effector memory T cells; TN, T naïve; TEMRA, terminally differentiated effector memory T cells re-expressing CD45RA; NKT, natural killer T cells; Breg, B regulatory cells.

## Data Availability

Not applicable.

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
