# Peer review of "Lymphocyte Subset Imbalance in Cardiometabolic Diseases: Are T Cells the Missing Link?"

_ijms, 2025, doi:10.3390/ijms26030868_

Round 1
Reviewer 1 Report (New Reviewer)
Comments and Suggestions for Authors
The manuscript aims to be a narrative review on the involvement of lymphocyte subsets in the cardiometabolic diseases, which present a public health problem. Although the research question is broad, this paper contains too extensive descriptions of the components of the immune system, methodologies in immunology, and disease entities. Thus, the structure of the scientific paper is completely lost. Particularly Introduction section should be focused, and Conclusion section should state the most important outcomes. The paper is lacking synthesis and rigour.
If prepared propely, this manuscript can be a valuable contribution to the literature.
1. The main question is the role of immunity/impaired lymphocyte subpopulations in the cardiometabolic (not cardiovascular as stated in the title, as diabetes is primarily a metabolic disorder) disease. Instead of clearly defined topics, the authors include overly extensive sections on immunity.
2. Again, the conclusions are too broad and too extensive, instead focused on the topic(s), the main questions are somehow immersed in the text; even the table is included.
3. The references are appropriate for the given manuscript, but far extensive for the specific research.
4. Too many tables and figures. Story-telling nature of teh article.
Comments on the Quality of English LanguageRequires editing.
Author Response
The manuscript aims to be a narrative review on the involvement of lymphocyte subsets in the cardiometabolic diseases, which present a public health problem. Although the research question is broad, this paper contains too extensive descriptions of the components of the immune system, methodologies in immunology, and disease entities. Thus, the structure of the scientific paper is completely lost. Particularly Introduction section should be focused, and Conclusion section should state the most important outcomes. The paper is lacking synthesis and rigour.
If prepared properly, this manuscript can be a valuable contribution to the literature.
Response to General Comments. We thank this Reviewer for suggested changes to our review, and we hope we have improved the organization of our work.
Comment 1. The main question is the role of immunity/impaired lymphocyte subpopulations in the cardiometabolic (not cardiovascular as stated in the title, as diabetes is primarily a metabolic disorder) disease. Instead of clearly defined topics, the authors include overly extensive sections on immunity.
Response to Comment 1. We thank the Reviewer for this point, and first we have changed the title to “Lymphocyte subsets imbalance in cardiometabolic diseases: are T cells the missing link?” and we have changed “Cardiovascular diseases (CVD)” to “Cardiovascular and cardiometabolic diseases (CVD)”.
We have shortened the introduction by reducing the immune system part, as follows.
“T and B lymphocytes, Natural Killer cells (NK), and innate lymphoid cells play pivotal roles in orchestrating immune responses and maintaining immune homeostasis in healthy conditions, as they possess distinct phenotypic and functional activities, enabling appropriate responses to antigens and pathogens [1,2]. Over recent decades, our knowledge on lymphocyte functions in immunity has significantly expanded, with innovative studies revealing their involvement in chronic inflammatory conditions, including cardiovascular and cardiometabolic diseases (CVD) [3]. T lymphocytes represent 10-25% of circulating peripheral blood cells and are mainly divided in two major groups of CD4+ and CD8+ αβ T cells, along with a small population of γδ T cells (1-5%) and natural killer T (NKT) cells (0.01-0.5%) [2,4]. CD4+ T helper (Th) cells can differentiate into different subsets, each expressing distinct surface molecules, cytokines, and transcription factors, thereby regulating immune responses in diverse ways [5]. For example, Th1 cells promote immune responses against intracellular bacteria and viruses by activating cytotoxic CD8+ T cells through the secretion of pro-inflammatory cytokines, such as interferon-gamma (IFN-γ) [6,7]. Similarly, B cells and innate lymphoid cells (ILCs) have attracted significant attention for their ability to shape inflammatory responses and influence tissue remodeling process in both physiological and pathological conditions [8].
Identification and characterization of lymphocyte subsets rely on standard methodologies, including cell sorting, and functional assays, and multiparametric flow cytometry (FCM) which is the most commonly used and widely employed technique [9]. Furthermore, FCM can be combined with cell sorting for specific cell subset isolation. Functional assays include measurements of cytokine levels, immune cell proliferation cytotoxic activity, other effector functions in response to specific stimuli [10]. Molecular biology techniques, such as polymerase chain reaction (PCR) and gene expression analysis, are used to further characterize lymphocyte subsets based on gene expression profiles, up- or downregulation of specific signaling pathways, or specific genetic polymorphisms, that might affect and alter immune responses [11].
CVD remains a leading cause of mortality worldwide, although increasing awareness to reduce risk factors and wide utilization of medications for primary prevention and disease treatment. Emerging evidence identifies the immune system as a crucial player in development and progression of all types of CVD, by regulating post-injury inflammation and tissue damage resolution through a fine-tuned crosstalk between immune cells, endothelial cells, cardiomyocytes, and fibroblasts. Indeed, exaggerated uncontrolled responses can either lead to excessive tissue damage or fibrosis [12]. This dual role of the immune system underscores an intricate involvement of these cells in the pathogenesis of CVD, still not fully understood. For example, CD4+ T cells are involved in inflammatory responses in atherosclerosis and hypertension, while CD8+ T cells are linked to tissue injury and repair mechanisms during myocardial infarction and stroke [65, 113]. These dynamics highlight the therapeutic potential of targeting T cell subsets in CVD. NKT cells could also modulate vascular inflammation and influence plaque composition; however, their exact roles are still under investigation, as well as how this regulation may shift in response to different stages of CVD progression [87]. Recent studies have emphasized the need for comprehensive immune profiling in patients with CVD to better delineate how different lymphocyte populations contribute to disease pathogenesis. The CVD immunome should include specific markers associated with immune cell activation, differentiation, and tissue infiltration, along with identification of molecular pathways that regulate immune cell responses in cardiovascular tissues (Figure 1) [12].
In addition, this intricate relationship between CVD and the immune system is further underscored by the increased risk of cardiovascular complications and mortality observed in autoimmune diseases and Acquired Immune Deficiency Syndrome (AIDS) [131-134]. Despite encouraging preclinical data, a substantial translational effort is required to bridge the gap between bench and bedside, and to realize comprehensive T cell characterization in CVD for future T cell-directed therapeutic strategy development.
In this review, we provide a comprehensive overview of lymphocyte contributions in CVD, with a focus on atherosclerosis, hypertension, diabetes, stroke, and myocardial diseases. We also summarize current FCM applications in characterization and monitoring of immune cell subsets in CVD, as potential diagnostic biomarkers and therapeutic targets. Flow cytometry immunophenotyping could represent in the near future a powerful diagnostic tool for enhancing clinical management of CVD patients, better disease monitoring, and could identify targetable cell populations with novel specific therapies. Indeed, bridging the gap between immunology and cardiology could represent the turning point for advancing our understanding of CVD and for fostering the development of more targeted and effective therapies for these patients..”
Comment 2. Again, the conclusions are too broad and too extensive, instead focused on the topic(s), the main questions are somehow immersed in the text; even the table is included.
Response to Comment 2. We thank the Reviewer for this suggestion, and we have shortened the conclusion section as follows.
“8. Conclusions
Comprehensive lymphocyte subset immunophenotyping in clinical settings represents a long-standing area of interest, yet its clinical application remains limited due to substantial technical and logistical challenges: (i) the rarity of certain subpopulations, that reduces sensitivity and specificity of FCM for their quantification, and requires optimized antibody panels, rigorous gating strategies, and large sample volumes, not always available, especially in pediatric or critically ill patients; (ii) inter-operator and inter-laboratory variabilities, such as sample handling, staining protocols, and instrument calibration, that introduce discrepancies thus reducing results comparability; (iii) pre-analytical variables, such as sample storage conditions and processing times, that add other variations affecting rare subpopulations quantification. Therefore, protocol standardization is essential to establish international consensus criteria for processing, analysis, and report for more reliable and comparable results, especially in clinical settings, including CVD.
Despite these challenges, integration of immune profiling into the diagnostic and therapeutic landscape of CVD holds great promise, as a better understanding of the intricate crosstalk between immune cells and disease pathology could enhance prognostic accuracy and the development of targeted therapies, such as selective modulation of Tregs. Furthermore, identification of immune-based biomarkers for early disease detection and therapy response prediction represents a promising frontier. While animal models have significantly contributed to advancing methodological approaches, their species-specific differences warrant careful extrapolation of findings to human disease. Consequently, refinement of cytometric techniques and the identification of biomarkers will be pivotal in overcoming current limitations and unlocking the full potential of lymphocyte subpopulation analysis for improving the clinical management of CVD.”
Comment 3. The references are appropriate for the given manuscript, but far extensive for the specific research.
Response to Comment 3. We understand this Reviewer’s point, and the number of references have been reduced.
Comment 4. Too many tables and figures. Story-telling nature of the article.
Response to Comment 4. We agree with this Reviewer’s comment, and we have reduced the number of tables to 2 and the number of figures to 2.
Reviewer 2 Report (Previous Reviewer 2)
Comments and Suggestions for Authors
This paper explores the role of lymphocytes in the development and progression of cardiovascular diseases (CVDs), highlighting the involvement of lymphocytes subsets. ​ It discusses how CD4 and CD8 T lymphocytes and NK cells contribute to inflammation, atherosclerosis, hypertension, diabetes, myocardial infarction, and stroke. ​ The review emphasizes the importance of understanding the function of these immune cells in CVD. ​ Overall, it calls for integrating immune profiling into clinical practice to enhance diagnostic and therapeutic strategies for CVDs. ​
This is a revised submission. My previous main comment was that it was somewhat frustrating to read. Although it has been improved, I cannot say it flows easily. The main structure of the paper needs major reorganization.
For instance, the reasons why the authors believe lymphocytes are of particular interest in CVD are scattered throughout the introduction, various main sections (atherosclerosis, hypertension, etc.), and the conclusions. The first paragraph, for example, moves too quickly to really convince readers that studying lymphocytes is crucial for advancing CVD prevention and treatment. This point is of major importance here, so the rationale must be much more supported and clear.
Additionally, although there have been improvements, some sections still need to clarify whether the authors are referring to human or animal (mostly mouse) studies. For instance, the authors added a large table about T cell markers, which I assume are for human cells. Do the clones presented in Table 2 recognize both human and mouse cells? There are numerous challenges in translating cell markers used for human studies to mouse studies and vice versa. While some elements of Table 2 may be helpful, it seems overly technical for the present review. I suggest presenting the table as supplementary information to improve the text's flow.
Although I support the idea of studying lymphocyte subpopulations in clinical settings, we are not the first to consider this. What are the challenges? Why has this not been better studied so far? As an example, the subpopulations mentioned sometimes represent less than 1% of lymphocytes, making their quantification and reproducibility in different lab settings challenging. These challenges may differ between human and mouse studies. The challenges should be clearly identified in the text.
Author Response
This paper explores the role of lymphocytes in the development and progression of cardiovascular diseases (CVDs), highlighting the involvement of lymphocytes subsets. It discusses how CD4 and CD8 T lymphocytes and NK cells contribute to inflammation, atherosclerosis, hypertension, diabetes, myocardial infarction, and stroke. The review emphasizes the importance of understanding the function of these immune cells in CVD. Overall, it calls for integrating immune profiling into clinical practice to enhance diagnostic and therapeutic strategies for CVDs.
Comment 1. This is a revised submission. My previous main comment was that it was somewhat frustrating to read. Although it has been improved, I cannot say it flows easily. The main structure of the paper needs major reorganization.
Response to Comment 1. We thank the Reviewer for recognizing the improvements after the first rounds of revisions. We hope we have made all the required changes in this revised work.
Comment 2. For instance, the reasons why the authors believe lymphocytes are of particular interest in CVD are scattered throughout the introduction, various main sections (atherosclerosis, hypertension, etc.), and the conclusions. The first paragraph, for example, moves too quickly to really convince readers that studying lymphocytes is crucial for advancing CVD prevention and treatment. This point is of major importance here, so the rationale must be much more supported and clear.
Response to Comment 2. We thank the Reviewer for this point, and we have made introduction section and the aim of our review clearer, as follows.
“T and B lymphocytes, Natural Killer cells (NK), and innate lymphoid cells play pivotal roles in orchestrating immune responses and maintaining immune homeostasis in healthy conditions, as they possess distinct phenotypic and functional activities, enabling appropriate responses to antigens and pathogens [1,2]. Over recent decades, our knowledge on lymphocyte functions in immunity has significantly expanded, with innovative studies revealing their involvement in chronic inflammatory conditions, including cardiovascular and cardiometabolic diseases (CVD) [3]. T lymphocytes represent 10-25% of circulating peripheral blood cells and are mainly divided in two major groups of CD4+ and CD8+ αβ T cells, along with a small population of γδ T cells (1-5%) and natural killer T (NKT) cells (0.01-0.5%) [2,4]. CD4+ T helper (Th) cells can differentiate into different subsets, each expressing distinct surface molecules, cytokines, and transcription factors, thereby regulating immune responses in diverse ways [5]. For example, Th1 cells promote immune responses against intracellular bacteria and viruses by activating cytotoxic CD8+ T cells through the secretion of pro-inflammatory cytokines, such as interferon-gamma (IFN-γ) [6,7]. Similarly, B cells and innate lymphoid cells (ILCs) have attracted significant attention for their ability to shape inflammatory responses and influence tissue remodeling process in both physiological and pathological conditions [8].
Identification and characterization of lymphocyte subsets rely on standard methodologies, including cell sorting, and functional assays, and multiparametric flow cytometry (FCM) which is the most commonly used and widely employed technique [9]. Furthermore, FCM can be combined with cell sorting for specific cell subset isolation. Functional assays include measurements of cytokine levels, immune cell proliferation cytotoxic activity, other effector functions in response to specific stimuli [10]. Molecular biology techniques, such as polymerase chain reaction (PCR) and gene expression analysis, are used to further characterize lymphocyte subsets based on gene expression profiles, up- or downregulation of specific signaling pathways, or specific genetic polymorphisms, that might affect and alter immune responses [11].
CVD remains a leading cause of mortality worldwide, although increasing awareness to reduce risk factors and wide utilization of medications for primary prevention and disease treatment. Emerging evidence identifies the immune system as a crucial player in development and progression of all types of CVD, by regulating post-injury inflammation and tissue damage resolution through a fine-tuned crosstalk between immune cells, endothelial cells, cardiomyocytes, and fibroblasts. Indeed, exaggerated uncontrolled responses can either lead to excessive tissue damage or fibrosis [12]. This dual role of the immune system underscores an intricate involvement of these cells in the pathogenesis of CVD, still not fully understood. For example, CD4+ T cells are involved in inflammatory responses in atherosclerosis and hypertension, while CD8+ T cells are linked to tissue injury and repair mechanisms during myocardial infarction and stroke [65, 113]. These dynamics highlight the therapeutic potential of targeting T cell subsets in CVD. NKT cells could also modulate vascular inflammation and influence plaque composition; however, their exact roles are still under investigation, as well as how this regulation may shift in response to different stages of CVD progression [87]. Recent studies have emphasized the need for comprehensive immune profiling in patients with CVD to better delineate how different lymphocyte populations contribute to disease pathogenesis. The CVD immunome should include specific markers associated with immune cell activation, differentiation, and tissue infiltration, along with identification of molecular pathways that regulate immune cell responses in cardiovascular tissues (Figure 1) [12].
In addition, this intricate relationship between CVD and the immune system is further underscored by the increased risk of cardiovascular complications and mortality observed in autoimmune diseases and Acquired Immune Deficiency Syndrome (AIDS) [131-134]. Despite encouraging preclinical data, a substantial translational effort is required to bridge the gap between bench and bedside, and to realize comprehensive T cell characterization in CVD for future T cell-directed therapeutic strategy development.
In this review, we provide a comprehensive overview of lymphocyte contributions in CVD, with a focus on atherosclerosis, hypertension, diabetes, stroke, and myocardial diseases. We also summarize current FCM applications in characterization and monitoring of immune cell subsets in CVD, as potential diagnostic biomarkers and therapeutic targets. Flow cytometry immunophenotyping could represent in the near future a powerful diagnostic tool for enhancing clinical management of CVD patients, better disease monitoring, and could identify targetable cell populations with novel specific therapies. Indeed, bridging the gap between immunology and cardiology could represent the turning point for advancing our understanding of CVD and for fostering the development of more targeted and effective therapies for these patients.”
Comment 3. Additionally, although there have been improvements, some sections still need to clarify whether the authors are referring to human or animal (mostly mouse) studies. For instance, the authors added a large table about T cell markers, which I assume are for human cells. Do the clones presented in Table 2 recognize both human and mouse cells? There are numerous challenges in translating cell markers used for human studies to mouse studies and vice versa. While some elements of Table 2 may be helpful, it seems overly technical for the present review. I suggest presenting the table as supplementary information to improve the text's flow.
Response to Comment 3. We thank the Reviewer for this point, and we can confirm that all chosen markers are for human T cell subsets immunophenotyping, and no OMIP panels referred to other animal species were chosen. The title of this table was changed as follows “Table 1. T cell markers and published standardized antibodies and gating strategies for human T cell subset immunophenotyping.”. We might agree with the Reviewer that this table is too technical; however, we would like to keep it in the main text, as the Academic Editor requested to focus more on flow cytometry.
Comment 4. Although I support the idea of studying lymphocyte subpopulations in clinical settings, we are not the first to consider this. What are the challenges? Why has this not been better studied so far? As an example, the subpopulations mentioned sometimes represent less than 1% of lymphocytes, making their quantification and reproducibility in different lab settings challenging. These challenges may differ between human and mouse studies. The challenges should be clearly identified in the text.
Response to Comment 4. We agree with the challenge raised by the Reviewer.
The investigation of lymphocyte subpopulations in clinical settings, while a long-standing area of interest, has not yet achieved widespread clinical application. This limited adoption stems from a series of substantial technical and logistical hurdles that impede accurate and reliable analysis. A primary challenge lies in the inherent rarity of certain lymphocyte subpopulations, which often constitute less than 1% of the total lymphocyte population. This low frequency has cascading effects on several aspects of the research process.
Firstly, precise quantification of these rare cells becomes exceedingly difficult. Highly sensitive and specific techniques are required, most notably multi-parameter flow cytometry. This technique necessitates the careful selection and optimization of antibody panels, coupled with stringent gating strategies to accurately identify and isolate the cells of interest. Even with these advanced tools, achieving statistically robust cell counts often demands large sample volumes. This requirement can be particularly problematic in clinical settings where sample availability may be limited, especially from pediatric or critically ill patients.
Secondly, the low abundance of these subpopulations significantly compromises the reproducibility of experiments, both between different laboratories and even within the same laboratory over time. Minor variations in a multitude of experimental parameters, such as sample handling procedures, staining protocols, instrument calibration, and data analysis methods, can introduce substantial discrepancies in the reported frequencies of these rare cell types. This lack of standardization poses a significant obstacle to comparing results across different studies and, crucially, hinders the translation of research findings into practical clinical applications.
Thirdly, the very methodologies employed to study these cells have inherent limitations. While flow cytometry remains a cornerstone technique, it is susceptible to technical artifacts such as spectral overlap between fluorophores, autofluorescence from cellular components, and issues related to cell viability. These factors can further confound the accurate identification and quantification of rare populations. Although more advanced techniques like mass cytometry (CyTOF) offer the advantage of higher dimensionality, allowing for the simultaneous analysis of a greater number of markers, they too have their limitations, including lower throughput and higher cost, which can restrict their widespread use.
Fourthly, the viability and phenotypic state of lymphocytes are highly sensitive to pre-analytical variables. These variables encompass factors such as the time elapsed between sample collection and processing, the temperature at which samples are stored, and the specific anticoagulants used during collection. These pre-analytical factors can introduce significant variability and bias into the results, particularly when dealing with rare subpopulations that are already challenging to detect and quantify.
Standardizing experimental protocols is paramount to establishing lymphocyte subpopulation analysis as a reliable and reproducible clinical test. This standardization, achieved through collaborative efforts across multiple laboratories, will enable the definition of robust cut-off values for healthy controls and patients with cardiovascular disease (CVD). This collaborative approach is essential for the widespread adoption of this analysis in routine clinical practice.
While mouse models are frequently used in immunological research and offer advantages such as reagent availability, validated antibody clones, and the availability of genetically modified strains, it is crucial to acknowledge the potential for species-specific differences. Extrapolating findings from mouse models to human disease requires careful consideration of potential discrepancies in immune cell development, function, and disease pathogenesis. Nevertheless, protocols developed in experimental animal models have significantly contributed to advancing the execution and analysis of cytometric data.
Although in the revised manuscript, we have explicitly reported these challenges, we believe that parallel efforts involving both human studies and animal models will be essential to overcome these challenges and drive further progress in the field.
This manuscript is a resubmission of an earlier submission. The following is a list of the peer review reports and author responses from that submission.
Round 1
Reviewer 1 Report
Comments and Suggestions for Authors
The idea of this review is very interesting. However, the article is lacking in many areas, making it unfit to be published. The information given in many parts of the article seems to lack focus. There are many filler sentences/phrases/paragraphs. As the line numbers are not present, I cannot give specific examples.The authors should try, ab initium, to establish a clear aim of the study, which should be more targeted, and then built upon it. As it written now, it is written more like to be a book chapter.
Another major issue relates to the used references, which are often not properly (or at all cited. Just as an example, the first chapter lacks any references, even if there are specific references to "recent studies"; according to the authors, they should have been gathered in Table 2, which also does not have references.
Figures 1 and 2 includes both some images and a Table, with a very low quality, potentially suggesting it being taken from someplace else? But no references are, again, used.
Author Response
General Comments. The idea of this review is very interesting. However, the article is lacking in many areas, making it unfit to be published. The information given in many parts of the article seems to lack focus. There are many filler sentences/phrases/paragraphs. As the line numbers are not present, I cannot give specific examples. The authors should try, ab initium, to establish a clear aim of the study, which should be more targeted, and then built upon it. As it written now, it is written more like to be a book chapter.
Response to General Comments. We thank the Reviewer for carefully reading our manuscript and for giving helpful comments and points of discussion. The revised version of the manuscript has been extensively revised, as suggested. We have refined the research question to focus selectively on the involvement of lymphocyte subsets in cardiovascular diseases. Indeed, we have changed the title as follows “Lymphocyte subset imbalance in cardiovascular diseases: are T cells the missing link?”, to better reflect core findings of our review. We also apologize for missing reference citations, and we have added, where appropriate, increasing the number of references from 68 to 104.
On page 3, lines 89-98, the aim of our review has been reorganized as follows “In this review, we provide a comprehensive overview of lymphocyte contributions in CVD, with a focus on atherosclerosis, hypertension, diabetes, stroke, and myocardial diseases. We also summarize current FCM applications in characterization and monitoring of immune cell subsets in CVD, as potential diagnostic biomarkers and therapeutic targets. Flow cytometry immunophenotyping could represent in the near future a powerful diagnostic tool for enhancing clinical management of CVD patients, better disease monitoring, and could identify targetable cell populations with novel specific therapies. Indeed, bridging the gap between immunology and cardiology could represent the turning point for advancing our understanding of CVD and for fostering the development of more targeted and effective therapies for these patients.”
Language has been revised to ensure that it is clear and concise, summarizing existing studies and indicating which areas require further investigation.
We believe that these revisions have significantly improved the quality and focus of the review article.
Comment 1. Another major issue relates to the used references, which are often not properly (or at all cited. Just as an example, the first chapter lacks any references, even if there are specific references to "recent studies"; according to the authors, they should have been gathered in Table 2, which also does not have references.
Response to Comment 1. We apologize again for missing reference citations, and we have added, where appropriate, increasing the number of references from 68 to 104. As suggested, we have included a new Table 2 with relevant studies, as displayed below.
Table 2. Species, diseases, and cell subset involvement in published literature.
Human |
|||
Disease |
Cell subset |
Tissues |
References |
Atherosclerosis |
CD8+ CD8+ TEMRA CD4+ |
Endomyocardial biopsy Blood Atherosclerotic plaque |
Friebel, J., et al [29]. Grivel, J.-C., et al.[31] Fan, L., et al. [34] |
Hypertension |
CD4+ and CD8+ |
Blood |
Itani, HA., et al [44] |
Diabetes |
CD4+ TEM CD4+ TN Treg NKT |
Blood |
Teniente-Serra, A., et al. [48] Nekoua, M.P., et al. [53] Jagannathan-Bogdan, M., et al. [54] Daryabor, G., et al. [56] |
Stroke |
CD4+ Treg |
Brain Blood |
Jin, W.N., et al. [67] Wang, H., et al. [72] |
Myocardial diseases |
CD8+ TEMRA CD4+ |
Blood Ventricular tissue |
Zhu, H.A.-O., et al. [82] Siamwala, J.A.-O., et al. [93] |
Mouse |
|||
Disease |
Cell subset |
Tissues |
References |
Atherosclerosis |
CD4+ TEMRA CD4+ CD8+ |
Heart cryosections Blood |
Delgobo, M., et al. [27] Zhou, X., et al [28] Padgett, L.E., et al.[32] |
Hypertension |
CD8+ |
Blood |
Hengel, F.E., J. P.Benitah, and U.O. Wenzel [43] |
Stroke |
CD4+ T cells |
Spleen Brain |
Jin, W.N., et al. [67] |
Myocardial diseases |
CD8+ TEMRA CD4+, CD8+ CD4+ CD19+CD5+ |
Blood Heart Pericardial adipose tissue |
Zhu, H.A.-O., et al.[82] Michel, Lars et al. [81] Siamwala, J.A.-O., et al.[93] Wu, L.A.-O.X., et al. [87] |
Rat |
|||
Disease |
Cell subset |
Tissues |
References |
Myocardial diseases |
Breg |
Hearts |
Huang, F., et al., [89] |
Abbreviations. Th, T helper; Tregs, T regulatory cells; DC, dendritic cell; CTLs, cytotoxic T cells; TEM, effector memory T cells; TCM, T central memory; TN, T naïve; TEMRA, Terminally Differentiated Effector Memory T Cells Re-expressing CD45RA; NK, Natural Killer; ILCs, innate lymphoid cells; Breg, B regulatory cells.
Comment 2. Figures 1 and 2 include both some images and a Table, with a very low quality, potentially suggesting it being taken from someplace else? But no references are, again, used.
Response to Comment 2. We apologize for missing references in Figures and Tables, which have been properly added. Please see updated version of Figures 1 and 2, and Table 1.
Reviewer 2 Report
Comments and Suggestions for Authors
This paper explores the role of the immune system in the development and progression of cardiovascular diseases (CVDs), highlighting the involvement of various leukocyte subsets. ​ It discusses how immune cells, particularly CD4 and CD8 T lymphocytes and NK cells contribute to inflammation, atherosclerosis, hypertension, diabetes, myocardial infarction, and stroke. ​ The review emphasizes the importance of understanding the function of these immune cells in CVD. ​ Overall, it calls for integrating immune profiling into clinical practice to enhance diagnostic and therapeutic strategies for CVDs.
​
While the paper covers important topics, it is somewhat frustrating to read. From the introduction, it is unclear why the authors focus specifically on T lymphocytes. More information on the research methodology would be beneficial. Additionally, it is often unclear whether the citations refer to human patients or mouse models. The title mentions “impaired subset distribution,” but the evidence supporting this claim is not well-explained. Are we looking at subset distribution in circulation or in tissues? More data from patients, as well as from DKO mouse models such as ApoE-/-CD4-/- or KO models affecting specific leukocyte subpopulations, would strengthen the paper. The focus on T lymphocytes alone could be sufficient and interesting, but the authors need to provide a more convincing argument about why we should not worry too much about neutrophils, monocytes and macrophages for instance.
The conclusions of the paper are not all strongly supported by the text.
Minor comments:
Tables 1 and 2 are difficult to read due to the lack of separation between rows. Improved formatting is needed.
For figures 1 and 2, it would be helpful to link the different parts of the tables to cited refs.
Author Response
This paper explores the role of the immune system in the development and progression of cardiovascular diseases (CVDs), highlighting the involvement of various leukocyte subsets. It discusses how immune cells, particularly CD4 and CD8 T lymphocytes and NK cells contribute to inflammation, atherosclerosis, hypertension, diabetes, myocardial infarction, and stroke. The review emphasizes the importance of understanding the function of these immune cells in CVD. Overall, it calls for integrating immune profiling into clinical practice to enhance diagnostic and therapeutic strategies for CVDs.
Response to General Comments. We thank the Reviewer for this positive feedback and for helpful comments that have markedly increased the quality of our work. We hope we have addressed all issues, as described below and wrote in red in the main text.
Comment 1. While the paper covers important topics, it is somewhat frustrating to read. From the introduction, it is unclear why the authors focus specifically on T lymphocytes. More information on the research methodology would be beneficial.
Response to Comment 1. We appreciate your suggestion to provide more context for our focus on lymphocyte subsets. The introduction section has been revised to more clearly articulate the rationale for focusing on lymphocyte subsets in cardiovascular diseases. We have highlighted the specific role of T cells in cardiovascular disease and emerging evidence supporting their involvement in its pathophysiology. We have changed the title as follows “Lymphocyte subsets imbalance in cardiovascular diseases: are T cells the missing link?”, to better reflect core findings of our review.
In addition, we have added a schematic figure summarizing T cell subsets involvement in the pathophysiology of cardiovascular disease in humans. We hope that these revisions have improved the clarity and comprehensiveness of the manuscript.
Comment 2. Additionally, it is often unclear whether the citations refer to human patients or mouse models. The title mentions “impaired subset distribution,” but the evidence supporting this claim is not well-explained.
Response to Comment 2. We thank the Reviewer for this valuable feedback, and we have clearly stated in the manuscript if studies have been conducted in human patients or in animal models. Moreover, for better clarity for the reader, we have added a new table (Table 2) with specific context of each finding and its relevance to human cardiovascular disease.
Please find below new Table 2.
Table 2. Species, diseases, and cell subset involvement in published literature.
Human |
|||
Disease |
Cell subset |
Tissues |
References |
Atherosclerosis |
CD8+ CD8+ TEMRA CD4+ |
Endomyocardial biopsy Blood Atherosclerotic plaque |
Friebel, J., et al [29]. Grivel, J.-C., et al.[31] Fan, L., et al. [34] |
Hypertension |
CD4+ and CD8+ |
Blood |
Itani, HA., et al [44] |
Diabetes |
CD4+ TEM CD4+ TN Treg NKT |
Blood |
Teniente-Serra, A., et al. [48] Nekoua, M.P., et al. [53] Jagannathan-Bogdan, M., et al. [54] Daryabor, G., et al. [56] |
Stroke |
CD4+ Treg |
Brain Blood |
Jin, W.N., et al. [67] Wang, H., et al. [72] |
Myocardial diseases |
CD8+ TEMRA CD4+ |
Blood Ventricular tissue |
Zhu, H.A.-O., et al. [82] Siamwala, J.A.-O., et al. [93] |
Mouse |
|||
Disease |
Cell subset |
Tissues |
References |
Atherosclerosis |
CD4+ TEMRA CD4+ CD8+ |
Heart cryosections Blood |
Delgobo, M., et al. [27] Zhou, X., et al [28] Padgett, L.E., et al.[32] |
Hypertension |
CD8+ |
Blood |
Hengel, F.E., J. P.Benitah, and U.O. Wenzel [43] |
Stroke |
CD4+ T cells |
Spleen Brain |
Jin, W.N., et al. [67] |
Myocardial diseases |
CD8+ TEMRA CD4+, CD8+ CD4+ CD19+CD5+ |
Blood Heart Pericardial adipose tissue |
Zhu, H.A.-O., et al.[82] Michel, Lars et al. [81] Siamwala, J.A.-O., et al.[93] Wu, L.A.-O.X., et al. [87] |
Rat |
|||
Disease |
Cell subset |
Tissues |
References |
Myocardial diseases |
Breg |
Hearts |
Huang, F., et al., [89] |
Abbreviations. Th, T helper; Tregs, T regulatory cells; DC, dendritic cell; CTLs, cytotoxic T cells; TEM, effector memory T cells; TCM, T central memory; TN, T naïve; TEMRA, Terminally Differentiated Effector Memory T Cells Re-expressing CD45RA; NK, Natural Killer; ILCs, innate lymphoid cells; Breg, B regulatory cells.
As suggested, we have revised the title to “Lymphocyte subsets imbalance in cardiovascular diseases: are T cells the missing link?”, to better reflect core findings of our review.
Comment 3. Are we looking at subset distribution in circulation or in tissues?
Response to Comment 3. We apologize for our misleading writing, and we have clarified this throughout the manuscript and in the new Table 2, where we have introduced the source of lymphocyte subsets in each study, specifying whether the lymphocyte dysregulation has been detected in peripheral blood or in specific tissues.
Comment 4. More data from patients, as well as from DKO mouse models such as ApoE-/-CD4-/- or KO models affecting specific leukocyte subpopulations, would strengthen the paper.
Response to Comment 4. We thank the Reviewer for this point and we have included additional data from mouse models on page 4, lines 148-150 “Furthermore, elevated frequencies of TSCM cells have been observed in Treg lineage tracker-ApoE−/− mice, suggesting that increased TSCM levels could represent potential hallmarks of advanced atherosclerosis in murine models [32].”
Comment 5. The focus on T lymphocytes alone could be sufficient and interesting, but the authors need to provide a more convincing argument about why we should not worry too much about neutrophils, monocytes and macrophages for instance.
Response to Comment 5. We thank the Reviewer for this valuable feedback and suggestions. We understand that our previous title was comprising of all leukocyte subsets, while we have not discussed them in the text. Therefore, as the principal focus of our review is on T lymphocytes, that offer a unique and complementary perspective in the pathophysiology of cardiovascular diseases, because of their antigen specificity and ability to modulate immune responses, offering a more targeted perspective on underlying disease mechanisms. Undoubtedly, other immune cells, such as neutrophils, monocytes, and macrophages, play a crucial role in early stages of the inflammatory response and in cellular debris clearance.
Comment 6. The conclusions of the paper are not all strongly supported by the text.
Response to Comment 6. We really thank the Reviewer for this careful review of our conclusions, that have been completely reorganized to ensure that they are strongly supported by the evidence presented in the text.
On pages 10-12, the following text was added.
“CVD remains a leading cause of mortality worldwide, although increasing awareness to reduce risk factors and wide utilization of medications for primary prevention and disease treatment. Emerging evidence identifies the immune system as a crucial player in development and progression of all types of CVD, by regulating post-injury inflammation and tissue damage resolution through a fine-tuned crosstalk between immune cells, endothelial cells, cardiomyocytes, and fibroblasts. Indeed, exaggerated uncontrolled responses can either lead to excessive tissue damage or fibrosis [97]. This dual role of the immune system underscores an intricate involvement of these cells in the pathogenesis of CVD, still not fully understood. Numerous studies have investigated the role of different lymphocyte subsets in CVD across various tissues and species, as detailed summarized in Table 2.
Within T cell subsets, CD4+ and CD8+ T cells, as well as Tregs, play a major role in modulating both inflammation and tissue remodeling. For example, CD4+ T cells are involved in inflammatory responses in atherosclerosis and hypertension, while CD8+ T cells are linked to tissue injury and repair mechanisms during myocardial infarction and stroke [34, 82]. Tregs, critical for maintaining immune balance and preventing fibrosis, often exhibit impaired functions in chronic inflammation, such as during atherosclerosis and heart failure [72]. These dynamics highlight the therapeutic potential of targeting T cell subsets in CVD. NKT cells could also modulate vascular inflammation and influence plaque composition; however, their exact roles are still under investigation, as well as how this regulation may shift in response to different stages of CVD progression [56]. Recent studies have emphasized the need for comprehensive immune profiling in patients with CVD to better delineate how different lymphocyte populations contribute to disease pathogenesis. The CVD immunome should include specific markers associated with immune cell activation, differentiation, and tissue infiltration, along with identification of molecular pathways that regulate immune cell responses in cardiovascular tissues (Figure 3) [12].
In this context, while other immune cells, including neutrophils, monocytes, and macrophages, are essential for the initial stages of the inflammatory response and for clearing cellular debris in the pathogenesis of cardiovascular diseases, we believe that T lymphocytes offer a distinctive and supplementary perspective [98]. Due to their antigen specificity and capacity to modulate the immune response over an extended period, they provide a more precise understanding of the mechanisms underlying cardiovascular diseases.
Clinical data supporting the key role of T-cell modulation in cardiovascular disease is still emerging. A randomized trial investigated canakinumab (IL-1β monoclonal antibody inhibiting the activity of Tregs) in patients with a history of myocardial infarction, demonstrating a significant reduction in the recurrence of cardiovascular events [92]. A randomized clinical trial in patients with stable ischemic heart disease and acute coronary syndrome using low-dose IL-2 employed Aldesleukin increased the average circulating Treg levels, alleviating endothelial dysfunction and reducing the formation of atherosclerotic plaques, favoring cardiovascular health [93, 94,98-100].
This intricate relationship between CVD and the immune system is further underscored by the increased risk of cardiovascular complications and mortality observed in autoimmune diseases and Acquired Immune Deficiency Syndrome (AIDS) [101-103]. For instance, in rheumatoid arthritis, the expansion of CD4+ and CD8+ TEM lymphocytes has been strongly associated with coronary artery calcification, suggesting a direct link between chronic inflammation and cardiovascular remodeling [103]. Similarly, during HIV infection, patients exhibit a significantly heightened risk of developing CVD, primarily due to microvascular dysfunction mediated by CD8+ PD1+ cells, elevated levels of pro-inflammatory cytokines such as TNF-α, high-sensitivity C-reactive protein, and dysregulated IL-6 signaling [104]. Despite encouraging preclinical data, a substantial translational effort is required to bridge the gap between bench and bedside, and to realize comprehensive T cell characterization in CVD for future T cell-directed therapeutic strategy development.
In conclusion, we believe that integrating detailed immune profiling, including the characterization of lymphocyte subsets and their functional states, into the diagnostic work-up and clinical management of CVD patients could significantly enhance our understanding of the disease biology and improve prognostic accuracy. Although much still needs to be done in terms of specific T-cell/pathology definition, we firmly believe that integrating the assessment of lymphocyte subpopulations into the clinical management of CVD could significantly contribute to the definition of this intricate crosstalk and offer promising opportunities for novel targeted therapies. By selectively modulating specific immune subsets—such as promoting the function of Tregs or inhibiting the inflammatory actions of T cells—these therapies could not only improve clinical outcomes but also reduce long-term disability and improve the quality of life for patients with CVD. Furthermore, the identification of immune-based biomarkers for early disease detection, as well as for predicting patient response to therapy, is an exciting frontier in the management of cardiovascular diseases.”
Minor comments:
Comment 7. Tables 1 and 2 are difficult to read due to the lack of separation between rows. Improved formatting is needed.
Response to Comment 7. We apologize for the lack of clarity of Tables 1 and 2. We have removed the previous table 2 with cell subsets, as requested, and we have introduced a new Table 2 with relevant studies. We have also added horizontal lines to separate different sections and enhance clarity. We hope these changes have significantly improved the visual presentation of our data and enhanced the overall reader experience.
Comment 8. For figures 1 and 2, it would be helpful to link the different parts of the tables to cited refs.
Response to Comment 8. We thank the Reviewer for highlighting figures issues. We have created both figures based on data and analysis collected and reported in the manuscript. In the revised version, we have replaced them with high-resolution versions. Moreover, to avoid any quality problems, a separate TIF version has been uploaded.
Round 2
Reviewer 1 Report
Comments and Suggestions for Authors
The authors have improved the quality of the article. As it is, it can be published.